# Additive effect of contrast and velocity suggests the role of strong excitatory drive in suppression of visual gamma response

Elena V. Orekhova[1,2,3]*, Andrey O. Prokofyev[1], Anastasia Yu. Nikolaeva[1], Justin F. Schneiderman[2,3], Tatiana A. Stroganova[1]

**1** Moscow State University of Psychology and Education, Center for Neurocognitive Research (MEG Center), Moscow, Russia, **2** University of Gothenburg, Sahlgrenska Academy, Institute of Neuroscience &Physiology, Department of Clinical Neuroscience, Gothenburg, Sweden, **3** MedTech West, Sahlgrenska University Hospital, Gothenburg, Sweden

* Orekhova.Elena.V@gmail.com

**Data Availability Statement:** All relevant data are within the paper and its Supporting Information files.

## Abstract

It is commonly acknowledged that gamma-band oscillations arise from interplay between neural excitation and inhibition; however, the neural mechanisms controlling the power of stimulus-induced gamma responses (GR) in the human brain remain poorly understood. A moderate increase in velocity of drifting gratings results in GR power enhancement, while increasing the velocity beyond some 'transition point' leads to GR power attenuation. We tested two alternative explanations for this nonlinear input-output dependency in the GR power. First, the GR power can be maximal at the preferable velocity/temporal frequency of motion-sensitive V1 neurons. This *'velocity tuning'* hypothesis predicts that lowering contrast either will not affect the transition point or shift it to a lower velocity. Second, the GR power attenuation at high velocities of visual motion can be caused by changes in excitation/inhibition balance with increasing excitatory drive. Since contrast and velocity both add to excitatory drive, this *'excitatory drive'* hypothesis predicts that the 'transition point' for low-contrast gratings would be reached at a higher velocity, as compared to high-contrast gratings. To test these alternatives, we recorded magnetoencephalography during presentation of low (50%) and high (100%) contrast gratings drifting at four velocities. We found that lowering contrast led to a highly reliable shift of the GR suppression transition point to higher velocities, thus supporting the excitatory drive hypothesis. No effects of contrast or velocity were found in the alpha-beta range. The results have implications for understanding the mechanisms of gamma oscillations and developing gamma-based biomarkers of disturbed excitation/inhibition balance in brain disorders.

## Introduction

Gamma-band oscillations arise from a precise interplay between neural excitation (*E*) and inhibition (*I*) [1, 2]. Neurochemical and optogenetic manipulations that change either *E* or *I* affect the amplitude and frequency of gamma oscillations [3–6], suggesting that gamma can

**Funding:** This work was supported by the Moscow State University of Psychology and Education; the Charity Foundation "Way Out"; Swedish Childhood Cancer (#MT2014-0007); Knut and Alice Wallenberg Foundation (#2014.0102) and Swedish Research Council (#2017-0068). The funders had no role in study design, data collection and analysis, decision to publish, or preparation of the manuscript.

**Competing interests:** The authors have declared that no competing interests exist.

provide useful biomarkers of an altered *E-I* balance in neuropsychiatric disorders [7]. Magnetoencephalographic (MEG) gamma oscillations induced in the visual cortex by large, moving, and high-contrast gratings (hereafter referred to as 'gamma response', GR) are particularly promising in this respect because they are individually stable [8] and genetically determined [9].

Gamma oscillations in the cerebral cortex are thought to be generated via interactions between excitatory pyramidal and inhibitory fast-spiking basket cells (pyramidal-interneuronal network gamma, i.e. PING)–this mechanism operates best in the 30–90 Hz regime [2, 10]. The frequency and power of visual gamma oscillations are strongly affected by sensory features of the visual stimuli. Understanding the mechanisms of such stimulus-dependency is crucial for potential use of the GR as a biomarker of *E-I* balance.

The GR peak frequency nearly linearly increases with luminance contrast [11–17] and drift-rate/velocity [18, 19] of a visual grating. The increase in GR frequency caused by increasing stimulation intensity is well explained by excitation of a network of interneurons [20]. Indeed, experimental studies suggest that tonic excitation of *I* interneurons is the major factor regulating gamma frequency [4, 21, 22].

Unlike that of frequency, intensity-related changes in the GR power are often not linear. In human MEG or EEG, GR power increases approximately linearly with increasing the gratings' contrast [11, 14, 23]. On the other hand, GR power recorded in monkey's local field potentials (LFP) saturates or even decreases at high contrasts [11, 12, 24]. Furthermore, increasing the motion-velocity/temporal frequency of a high contrast grating leads to bell-shaped changes in GR power in both human MEG [18] and in monkeys' LFP [19], with the maximal response being observed at around 1.2˚/s (or 2 Hz temporal frequency). There are at least two plausible explanations of this bell-shaped dependency of the MEG GR power on visual motion velocity.

*First*, this bell-shaped dependency may reflect the number of responsive V1 neurons involved in the generation of gamma activity. Indeed, positive correlations between neuronal spiking rate and LFP gamma power were observed in a number of studies [17, 25–28]; but see [12, 19]. In terms of spiking rate, neurons in V1 are 'tuned' to specific features of visual stimulation, such as orientation, spatial frequency, and temporal frequency/velocity [29]. Therefore, the stimuli that activate more neurons can potentially cause stronger gamma activity. For example, such spatial frequency tuning of neurons' firing rate in the primate cortical area V1 [30] has been invoked to explain the spatial frequency dependency of human visual gamma oscillations [31]. In a similar way, neurons in V1 and/or lateral geniculate nuclei (LGN) are preferentially tuned to a certain temporal frequency/velocity [30, 32–34]. This can manifest itself as a velocity tuning of visual gamma oscillations, which would result in the observed bell-shaped dependency of GR power on visual motion velocity. Neural selectivity for certain velocities or temporal frequencies of visual motion has furthermore been reported in studies that used other neuroimaging methods and stimulation parameters [35–38].

There are, however, some observations that are difficult to explain using the 'velocity tuning' hypothesis. Depending on the imaging methods used, the temporal frequencies that induce maximal response may differ [35–38]. Neural spiking rates furthermore do not necessarily peak at the same temporal frequencies as GR power does. For example, Salekhar et al recorded LFP and multi-unit activity (MUA) in monkeys' V1 in response to high-contrast gratings drifting at different temporal frequencies [19]. They report the bell-shaped dependency on the drift rate (temporal frequency) in both LFP gamma power (50–80 Hz) and MUA. However, the temporal frequency that induced the maximal neural spiking rate did not correspond to that of the maximal GR power (8–16 Hz for spiking, 1–4 Hz for GR power). In other words, increasing the temporal frequency of visual stimuli from 4 to 8 Hz induced an increase in the neural firing rate whereas GR power started to reduce. Considering such

observations, the factors other than 'velocity tuning' may contribute to non-linear dependency of gamma oscillation on velocity of visual gratings.

*The second*, alternative explanation for the bell-shaped velocity-related changes of the visual GR power is their dependency on excitatory drive. In order for the gamma rhythm to be recorded in LFP or MEG, it is necessary that the membrane potentials of a large number of principal neurons fluctuate in synchrony [39]. Such synchronous fluctuations are driven by synchronous perisomatic inhibitory post-synaptic potentials (IPSP) paced by parvalbumin-positive (PV) fast-firing interneurons [40, 41]. Modeling and experimental results suggest that a tightly maintained balance between synaptic *E* and *I* is critical for neural synchronization [41]. The *E/I* ratio in visual cortical networks is not fixed, but changes as a function of excit-atory drive [42]. For example, by using voltage-clamp measurements in mice, Adesnik et al have shown that an increase in stimulation intensity (i.e. contrast of a stationary grating) enhances both synaptic *E* and *I*, but concomitantly decreases the *E/I* ratio due to a steeper increase in *I* as compared to *E*. A further increase in excitatory drive (e.g., via adding motion to a high-contrast grating) might lead to even greater increase of *I* and progressively lower *E/I* ratio. There is then evidence that disproportionally strong *I* may disrupt synchronization in the gamma range [4, 10, 43–46]. It is, therefore, likely that strong *I* caused by strong excitatory drive may contribute to the observed nonlinear velocity-related changes in the GR power.

To test these hypotheses we used high-contrast (100%) and low-contrast (50%) gratings that moved at the same range of velocities. If the 'velocity-tuning hypothesis' is correct, then the maximal GR at lower contrast will be observed at the same or *lower* motion velocity com-pared to that found for the higher contrast. The latter can occur according to this hypothesis because a decrease in the stimulus contrast changes the tuning of the motion-sensitive V1 neu-rons to lower temporal frequencies [32, 47, 48]. On the other hand, an additive effect of con-comitant increases in contrast and velocity would favor the 'excitatory drive hypothesis'. The latter implies that the critical level of excitatory drive leading to the 'gamma response maxi-mum' would be reached at higher velocity at the low contrast, as compared to the high. If true, the link between strong excitatory drive and GR attenuation at high stimulation intensities may have important functional implications: individual variations in velocity-related attenua-tion of GR may reflect the capacity for inhibitory circuitry to down-regulate excessive *E* in the entire V1 network.

The changes in power at the high-frequency part of the neural response spectra are often accompanied by changes in neural oscillatory power at low-frequencies in the opposite direc-tion [49, 50]. To test for the specificity of the velocity-related changes in the gamma range, we analyzed effects of contrast and velocity also in the alpha-beta frequency range.

## Methods

### Participants

Seventeen neurologically healthy subjects (age 18–39, mean = 27.2, sd = 6.1; 7 males) were recruited for the study by advertising among students and staff of the Moscow State University of Psychology and Education (MSUPE). All participants had normal or corrected to normal vision. None of the participants reported the presence of any psychiatric problems. An informed consent form was obtained from all the participants. The study has been approved by the ethical committee of MSUPE.

### Experimental task

The visual stimuli were generated using Presentation software (Neurobehavioral Systems Inc., USA). We used a PT-D7700E-K DLP projector to present images with a 1280 x 1024 screen

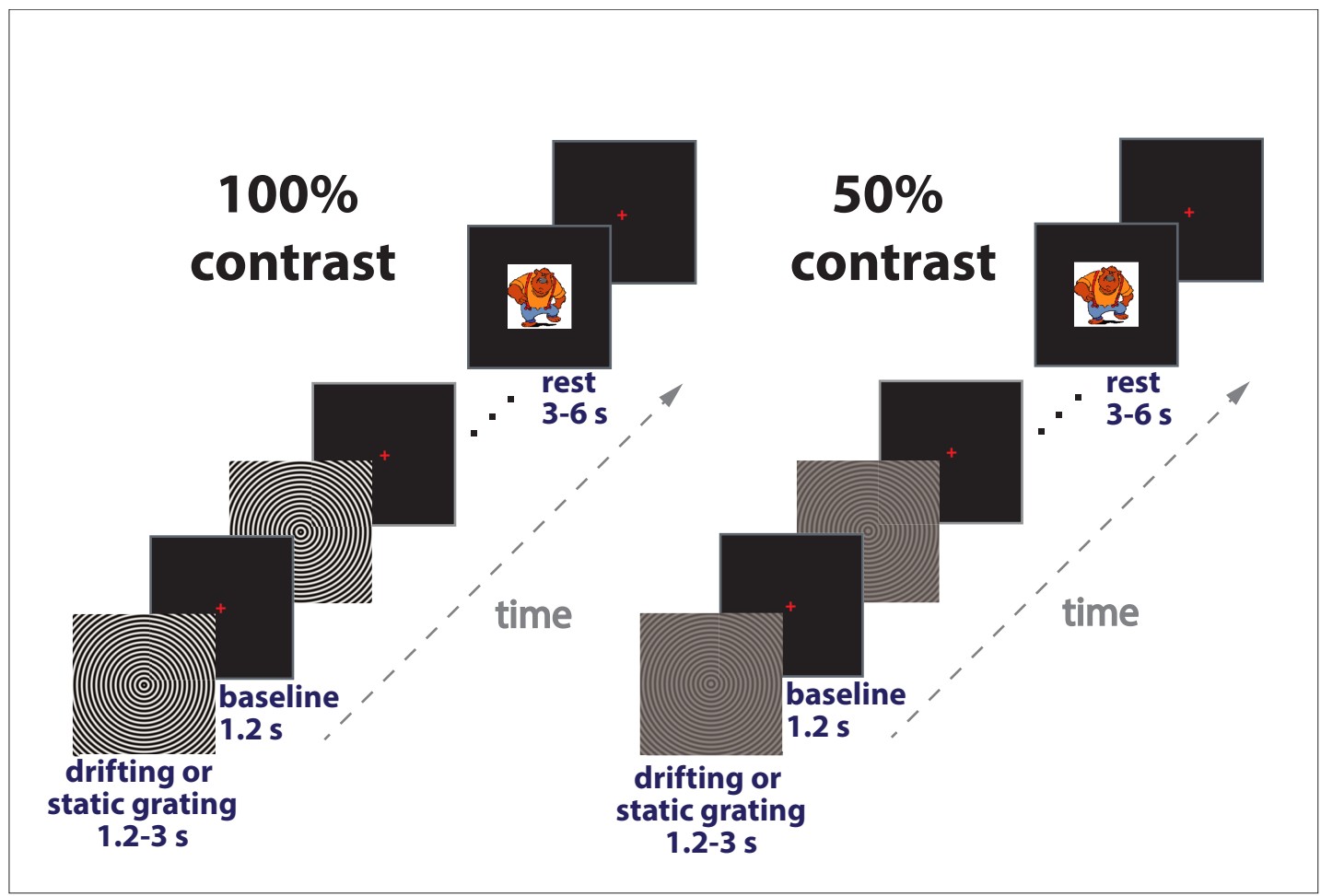

**Fig 1. Experimental design.** Each trial began with the presentation of a fixation cross that was followed by an annular grating that remained static (0˚/s) or drifted inward for 1.2–3 s at one of the three velocities: 1.2, 3.6, 6.0˚/s. The 100% and 50% contrast gratings were presented in separate sessions. Participants responded to the change in the stimulation flow (disappearance of the static grating or termination of motion) with a button press. Short animated cartoon characters were presented randomly between every 2–5 stimuli to maintain vigilance and reduce visual fatigue.

resolution and a 60 Hz refresh rate. The experimental paradigm is schematically presented in Fig 1. The stimuli were gray-scale sinusoidally modulated annular gratings presented with Michelson contrast of ~100% or 50%. Luminance of the display, measured from the eye position, was 46 Lux during presentation of both 50% and 100% contrast stimuli and 2 Lux during inter-stimulus intervals. The gratings had a spatial frequency of 1.66 cycles per degree of visual angle and covered $18 \times 18$ degrees of visual angle. They appeared in the center of the screen over a black background and drifted to the center of the screen immediately upon display with one of four velocities: 0, 1.2, 3.6, or 6.0˚/s, (0, 2, 6, or 10 Hz temporal frequency) referred to as 'static', 'slow', 'medium', and 'fast'. Each trial began with a 1200 ms presentation of a red fixation cross in the center of the display over a black background that was followed by the presentation of the grating that either remained static or drifted with one of the three velocities. After a randomly selected period of 1200–3000 ms, the movement stopped or the static stimulus disappeared. To keep participants alert, we asked them to respond to the change in the stimulation flow (stop of the motion or disappearance of the static stimulus) with a button press. If no response occurred within 1 second, a discouraging message, "too late!" appeared and remained

on the screen for 2000 ms, after which a new trial began. The static stimulus and those drifting at different rates were intermixed and appeared in a random order within each of three experimental blocks. The participants responded with either the right or the left hand in a sequence that was counterbalanced between blocks and participants. Each of the stimuli types was presented 30 times within each experimental block. In order to minimize visual fatigue and boredom, short (3–6 s) animated cartoon characters were presented between every 2–5 stimuli. The high (100%) and low (50%) contrast gratings were presented in different experimental sessions and the order of these sessions was counterbalanced between subjects.

## MEG data recording

Neuromagnetic brain activity was recorded using a 306-channel detector array (Vectorview; Neuromag, Helsinki, Finland). The subjects' head positions were continuously monitored during MEG recordings. Four electro-oculogram (EOG) electrodes were used to record horizontal and vertical eye movements. EOG electrodes were placed at the outer canti of the eyes and above and below the left eye. To monitor heartbeats, electrocardiogram (ECG) electrodes were placed at the manubrium sterni and the mid-axillary line (V6 ECG lead). MEG, EOG, and ECG signals were recorded with a band-pass filter of 0.03–330 Hz, digitized at 1 000 Hz, and stored for off-line analysis.

## MEG data preprocessing

The data was first de-noised using the Temporal Signal-Space Separation (tSSS) method [51] implemented in MaxFilter™ (v2.2) with parameters: '-st' = 4 and '-corr' = 0.90. For all three experimental blocks, the head origin position was adjusted to a common standard position. For further pre-processing, we used the MNE-python toolbox [52] as well as custom Python and MATLAB® (TheMathWorks, Natick, MA) scripts.

The de-noised data was filtered between 1 and 200 Hz and down sampled to 500 Hz. A discrete Fourier transform filter was applied to remove power-line noise (50 and 100 Hz). To remove biological artifacts (blinks, heart beats, and in some cases myogenic activity), we then applied independent component analysis (ICA). MEG signal with too high (4e-10 fT/cm for gradiometers and 4e-12 fT for magnetometers) or too low (1e-13 fT/cm for gradiometers and 1e-13 fT for magnetometers) amplitudes were excluded from the analysis. The number of independent components was set to the dimensionality of the raw 'SSS-processed' data (usually around 70). We further used an automated MNE-python procedure to detect EOG and ECG components, which we complemented with visual inspection of the ICA results. The number of rejected artifact components was usually 1–2 for vertical eye movements, 0–3 for cardiac, and 0–6 for myogenic artifacts.

The ICA-corrected data was then epoched from −1 to 1.2 sec relative to the stimulus onset. We then performed time-frequency multitaper analysis (2.5 Hz step; number of cycles = frequency/2.5) and excluded epochs contaminated by strong muscle artifacts via thresholding the high-frequency (70–122.5 Hz) power. For each epoch, the 70–122.5 Hz power was averaged over sensors and time points and the threshold was set at 3 standard deviations of this value. The remaining epochs were visually inspected for the presence of undetected high-amplitude bursts of myogenic activity and those contaminated by such artifacts were manually marked and excluded from the analysis. After rejection of artifacts, the average number of epochs per subject at 100% contrast was 79, 78, 78, and 79 for the 'static', 'slow', 'medium' and 'fast' conditions, respectively. For the 50% contrast, the respective values were 79, 80, 80, and 82.

## Structural MRI

Structural brain MRIs (1 mm$^3$ T1-weighted) were obtained for all participants and used for source reconstruction.

## Time-frequency analysis of the MEG data

The following steps of the data analyses were performed using Fieldtrip Toolbox functions (http://fieldtrip.fcdonders.nl; [53]) and custom scripts developed within MATLAB.

In order to decrease the contribution of phase-locked activity related to appearance of the stimulus on the screen and 60 Hz refresh rate of the projector, we subtracted the average evoked response from each data epoch. The lead field was calculated using individual 'single shell' head models and cubic 6 mm-spaced grids linearly warped to the MNI-atlas-based template grid. Time-frequency analysis of the MEG data was then performed in the following two steps.

*First*, we tested for the presence of reliable clusters of gamma enhancement and alpha suppression at the source level using the DICS inverse-solution algorithm [54] and the cluster-based permutation test [55]. The frequency of interest was defined individually for each subject and condition based on the sensor data (only the data from gradiometers were used at this stage). For the gamma range (45–90 Hz), we performed time-frequency decomposition of prestimulus ('pre': -0.9 to 0 s) and post-stimulus ('post': 0.3 to 1,2 s) MEG signals using 8 discrete prolate spheroidal sequences (DPSS) tapers with ±5 Hz spectral smoothing. For the alpha-beta range (7–15 Hz), we used 2 DPSS tapers and ±2 Hz spectral smoothing. We then calculated the average (post-pre)/pre ratio and found the 4 posterior sensors with the maximal post-stimulus increase in gamma power and those with the maximal post-stimulus power decrease in the alpha-beta band. By averaging those 4 channels, we identified the frequencies corresponding to the maximal post-stimulus power increase (for gamma) or decrease (for alpha-beta). The frequency ranges of interest were then established within 35–110 Hz (gamma) and 5–20 Hz (alpha-beta) limits, where the (post-pre)/pre ratios exceeded +/- 2/3 of the peak value (positive for gamma, negative for alpha-beta). The center of gravity of the power over these frequency ranges was used as the peak frequency for each band. Time-frequency decomposition was then repeated while centered at these peak frequencies. Common source analysis filters were derived for combined pre- and post-stimulus intervals using DICS beamforming with a 5% lambda parameter and fixed dipole orientation (i.e, only the largest of the three dipole directions per spatial filter was kept). The filter was then applied separately to the pre- and post-stimulus signals. Subsequently, we calculated univariate probabilities of pre- to post-stimulus differences in single trial power for each voxel using t-statistics. We then performed bootstrap resampling (with 10 000 Monte Carlo repetitions) between 'pre' and 'post' time windows to determine individual participants' maximal source statistics based on the sum of t-values in each cluster.

*Second*, in order to analyze response parameters at the source maximum, we used linearly constrained minimum variance (LCMV) beamformers [56]. The spatial filters were computed based on the covariance matrix obtained from the whole epoch and for the three experimental conditions with a lambda parameter of 5%. Prior to beamforming, the MEG signal was bandpass filtered between 30 and 120 Hz for the gamma range and low-pass filtered below 40 Hz for the alpha-beta range. The 'virtual sensors' time series were extracted for each brain voxel and time-frequency analysis with DPSS multitapers (~1 Hz frequency resolution) was performed on the virtual sensor signals. For the gamma range, we used ±5 Hz smoothing. For the alpha-beta range, this parameter was ±2 Hz. The *'maximally induced'* gamma voxel was defined as the voxel with the highest relative post-stimulus increase of 45–90 Hz power in the

'slow' motion velocity condition within the visual cortical areas (i.e. L/R cuneus, lingual, occipital superior, middle occipital, inferior occipital, and calcarine areas according to the AAL atlas [57]). The 'slow' condition was selected because the reliability of the GR was highest in this condition. The *'maximally suppressed'* voxel in the alpha-beta range was then defined as the voxel with the strongest relative suppression of 7–15 Hz power. The weighted peak parameters for the GR and the low-frequency response were calculated from the average spectra of the virtual sensors in 26 voxels closest to, and including, the 'maximally induced' and 'maximally suppressed' voxels, respectively, using the approach described for the sensor level analysis. For each individual and condition, we also assessed the reliability of the pre- to post-stimulus power changes at these 26 averaged voxels. The change was considered reliable if its absolute peak value was significant with $p < 0.0001$ (Wilcoxon signed rank test). The coordinates of the voxels demonstrating the highest power changes were defined in MNI coordinates (S1 Tables A and B in Supporting Information).

### Suppression transition velocity and frequency

We have previously shown that GR power initially increases and then decreases as a function of velocity of a high-contrast visual grating [18]. For each subject, we approximated the visual motion velocity at which GR power was maximal—i.e., the 'suppression transition velocity' (STVel), as the centre of gravity of the GR power as a function of visual motion velocity:

$$\text{STVel} = (\text{Pow}_0 {}^*0 + \text{Pow}_{1.2} {}^*1.2 + \text{Pow}_{3.6} {}^*3.6 + \text{Pow}_{6.0} {}^*6.0)/(\text{Pow}_0 + \text{Pow}_{1.2} + \text{Pow}_{3.6} + \text{Pow}_{6.0}),$$

where 'Pow' is the center of gravity GR peak power in the respective velocity condition (i.e. 0, 1.2, 3.6 or 6.0˚/s). Although the STVel may not precisely correspond to the velocity for which the GR was at maximum, it reflects the distribution of power between velocity conditions.

In a similar way, we approximated the GR frequency of the maximal GR–i.e. the suppression transition frequency (STFreq)—as the center of gravity of the GR power curve as a function of GR peak frequency for each subject (i.e., in the same way as we did for the STVel):

$$\text{STFreq} = (\text{Pow}_0 {}^*\text{Freq}_0 + \text{Pow}_{1.2} {}^*\text{Freq}_{1.2} + \text{Pow}_{3.6} {}^*\text{Freq}_{3.6} + \text{Pow}_{6.0} {}^*\text{Freq}_{6.0})/(\text{Pow}_0 + \text{Pow}_{1.2} + \text{Pow}_{3.6} + \text{Pow}_{6.0}),$$

where 'Freq' and 'Pow' are the peak GR frequency and power in the respective velocity conditions.

### Evaluation of rank-order consistency

To evaluate rank-order consistency of GR parameters at different grating velocities and contrasts, we calculated two types of correlations: 1) between contrasts for the respective velocity conditions and 2) between velocities, separately for each of the contrasts. Since frequency, but not power, of the visual gamma oscillations is strongly affected by age in adults [18, 58], we calculated its partial correlations with frequency, while using age as a nuisance variable.

## Results

### Gamma frequency range

**Localization and reliability.** Fig 2 shows group probability maps of the source localization of the induced GR.

For the 100% contrast stimuli, significant ($p < 0.05$) activation clusters with maxima in visual cortical areas were found in all participants and motion velocity conditions. For the 50%

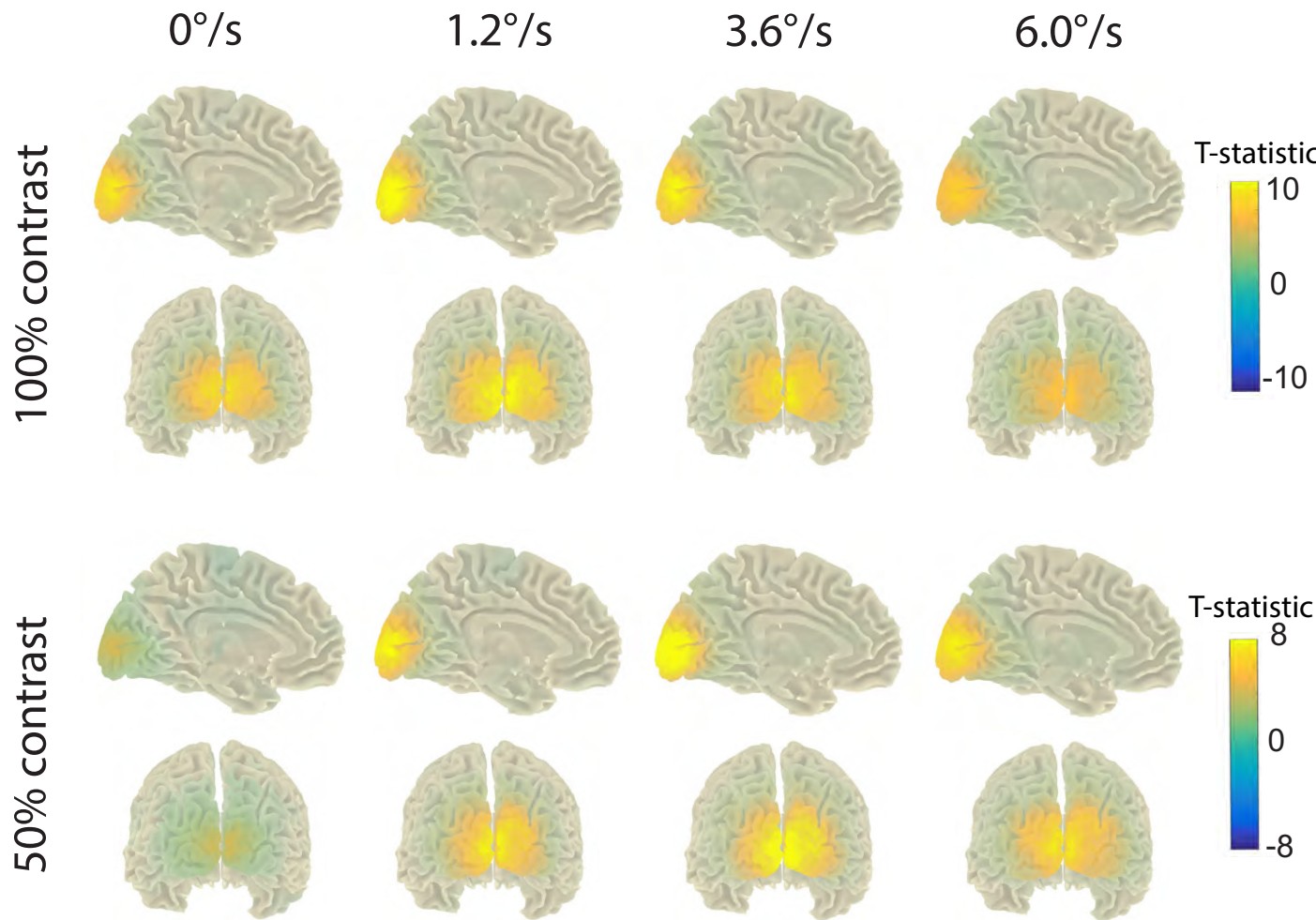

**Fig 2. Grand average statistical maps of the cortical GRs to drifting visual gratings.** The maps are given for the weighted peak gamma power, separately for the two contrasts and four motion velocities. Positive signs of the t-statistics correspond to stimulation-related increases in gamma power. Note the different scales for the 50% and 100% contrasts and that the magnitude of the GR was affected by both contrast and velocity.

contrast, significant activation clusters were absent (p>0.0001, see Methods for details) in one participant for all velocity conditions (A007), in two participants for the 'static' condition (A008 and A016), and in one participant for the 'fast' (6.0˚/s) condition (A008). The grand average and individual GR spectra at the 'maximally induced' group of voxels are shown in Figs 3 and 4. Aside from the exceptions listed above, increases in gamma power at the selected voxels were significant in all subjects and conditions according to our criteria.

The peaks that were unreliable according to at least one of the criteria (i.e. either the absence of an activated cluster or low probability (p>0.0001) of gamma power increase at the selected voxels) were further excluded from analyses of the peak frequency and the position of the maximally induced voxel, but not from analysis of the GR power. Note that exclusion of the unreliable GR peaks affected the degrees of freedom in the corresponding ANOVAs presented below.

In the majority of cases, the voxel with the maximal increase in gamma power was located in the calcarine sulcus (see S1A and S1B Table and for coordinates of the 'maximally induced'

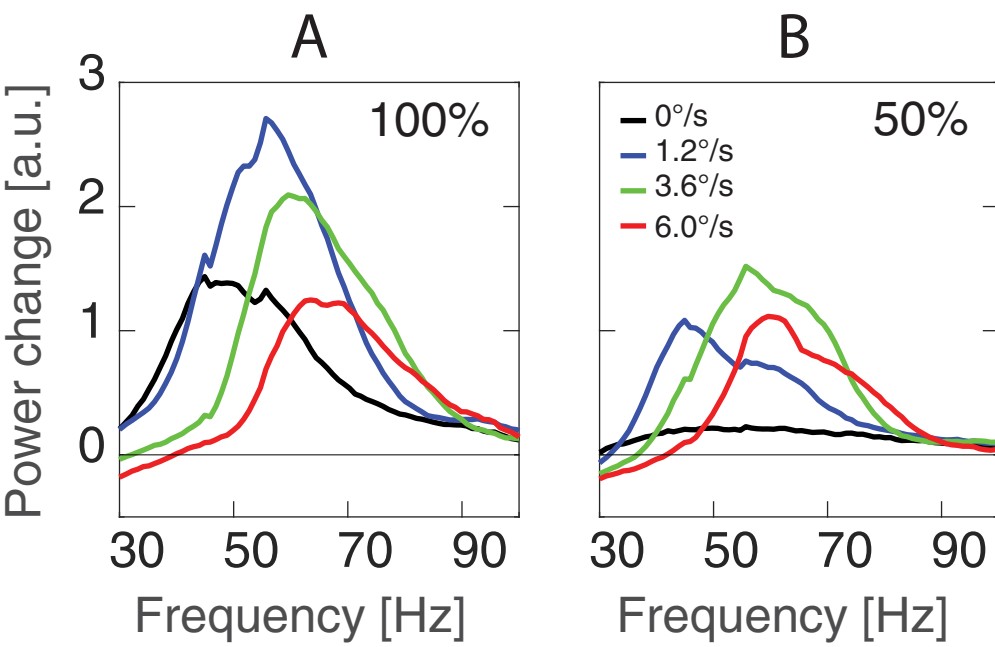

**Fig 3.** Grand average spectra of cortical GRs to the gratings of 100% (A) and 50% (B) contrasts drifting at four motion velocities. Here and hereafter, the GR power change is calculated as (post-pre)/pre ratio, where 'pre' and 'post' are the spectral power values in the -0.9 to 0 and 0.3 to 1.2 s time windows relative to the stimulus onset.

voxel). There was a significant effect of Velocity on the 'z' coordinates ($F_{(3,39)}$ = 4.7, epsilon = 0.84, p<0.05). In the 'fast' condition, the voxel with the maximal increase in gamma power was located on average 2.6 mm more superior relative to the other 3 conditions (MNI Z coordinate for the static: 6.1 mm, slow: 5.4 mm, medium: 5.2 mm, fast: 8.2 cm). There was also a Velocity*Contrast interaction for the 'y' coordinate ($F_{(3,39)}$ = 3.6, epsilon = 0.59, p<0.05), which is explained by a relatively more posterior source location of the gamma maximum in the 'fast' and low-contrast condition than in the 'fast' and high-contrast condition (-95 vs -92 mm). Considering the small condition-related differences in the position of the voxel with the maximal increase in gamma power (<6 mm, which was the size of the voxel used for the source analysis), all the stimuli activated largely overlapping parts of the primary visual cortex.

### Effects of contrast and velocity

**Frequency:** Fig 5A shows group mean GR peak frequency values for the two contrast and four velocity conditions. The GR peak frequency was strongly affected by the motion velocity of the grating ($F_{(3,39)}$ = 78.2, epsilon = 0.62, p<1e-6) and, to a lesser extent, by its contrast ($F_{(1,13)}$ = 14.3, p = 0.0023). Inspection of Fig 5A shows that an increase in velocity resulted in a substantial increase of the GR peak frequency. For the full-contrast grating, the peak frequency increased from 51.6 Hz ('static') to 68.6 Hz ('fast'). For the 50% contrast, the respective values were 51.1 and 63.0 Hz. Increasing contrast, on the other hand, led to a 5Hz increase in the peak frequency of the GR to moving stimuli. The distributions of individual peak frequencies are shown in S2 Fig in Supporting Materials.

**Power:** Fig 5B shows group mean GR peak power values for the two contrast and four velocity conditions. There was a highly significant effect of Contrast ($F_{(1,16)}$ = 66.2, p<1e-6) on GR peak power, which is explained by a generally higher GR power in case of high, as compared to low, contrast. The significant effect of Velocity ($F_{(3,48)}$ = 9.3, epsilon = 0.46, corrected

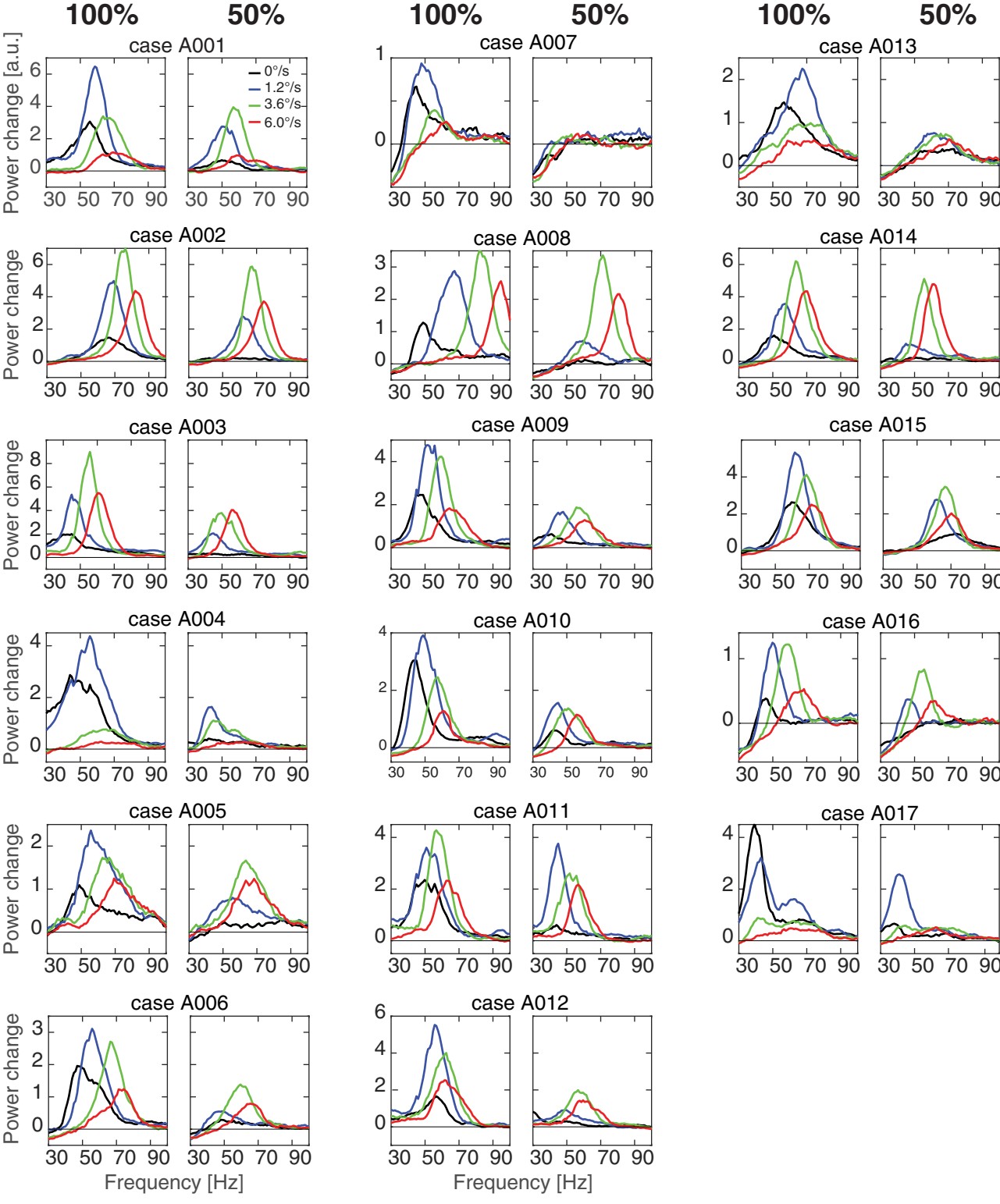

**Fig 4. Individual spectra of cortical GRs.** The plots are the same as in Fig 3, see Methods for further details.

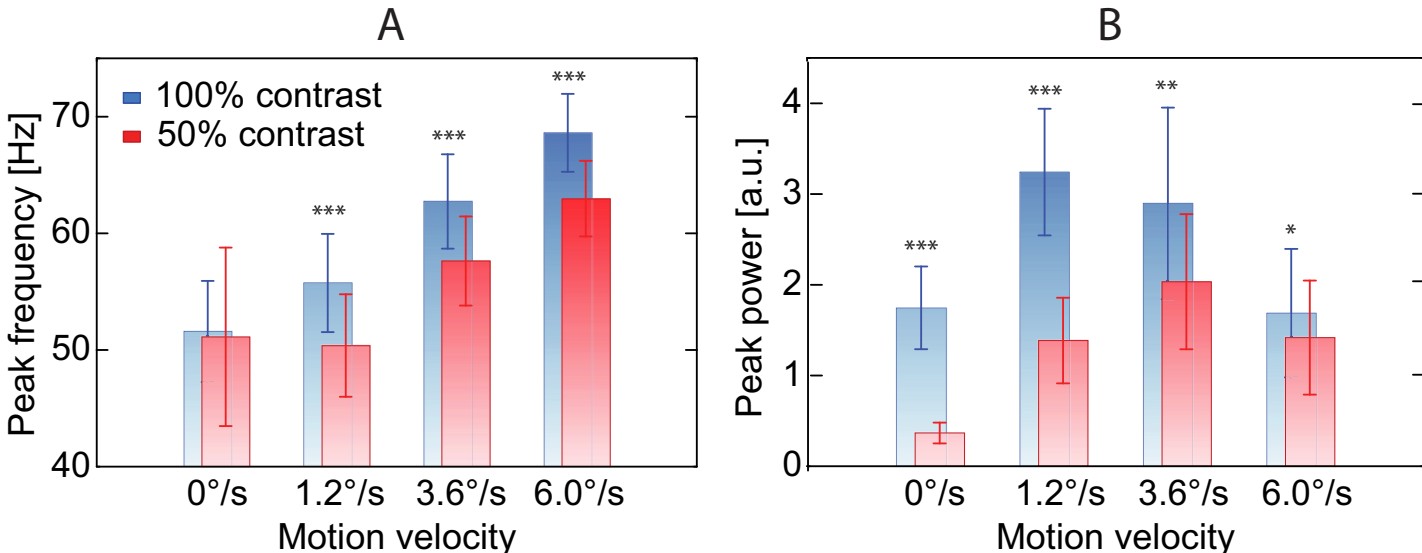

**Fig 5.** Grand average peak frequency (A) and power (B) of cortical GRs as a function of Contrast and motion Velocity. Vertical bars denote 0.95 confidence intervals. *** $p < 0.001$; ** $p < 0.01$; * $p < 0.05$.

$p = 0.003$) is due to an initial increase in GR power from the static to slow condition ($F_{(1,16)} = 43.1$, $p < 1e-5$) followed by a decrease from the medium to the fast velocity condition ($F_{(1,16)} = 14.2$, $p < 1e-4$). Importantly, there was highly significant Contrast*Velocity interaction effect ($F_{(3,48)} = 12.4$, epsilon = 0.78, corrected $p < 1e-4$). Contrast had a stronger effect on GR power for the static and slowly moving stimuli, as compared to the faster (medium and fast) velocities. As a result, the maximum of the inverted bell-shaped power vs. motion velocity curve moved to higher velocity in the low contrast condition, as compared to the full contrast one.

Generally, we confirmed our previous finding [18] of an inverted bell-shaped dependency of GR power on the motion velocity of full-contrast gratings and found the similar bell-shaped dependency for the gratings of 50% contrast. The significant Contrast*Velocity interaction showed that the form of this bell-shaped curve was modulated by the luminance contrast in such a way that the transition to GR suppression at low contrast occurred at a higher velocity.

**Suppression transition.** We further sought to investigate whether the shift to higher velocities in the GR suppression transition point under low contrast is robust enough to be detected at the individual subject level. For each subject, we calculated the 'suppression transition velocity' (STVel) as described in the Methods. Importantly, the STVel measure allows us to characterize contrast-dependent shifts in a more rigorous manner than relying exclusively on the discrete values of motion velocities for which the GR was maximal. Decreasing luminance contrast resulted in a reliable increase in STVel, and thus the GR suppression transition point, in all 17 participants ($F_{(1,16)} = 202.0$; $p < 1e-9$; Fig 6A).

Lowering contrast led to slowing of gamma oscillations; we therefore also tested if, irrespective of contrast, the point of transition to GR suppression corresponds to a certain GR frequency. To do this we calculated the individual 'suppression transition frequencies' (STFreq) as described in the Methods. The STFreq was slightly, but significantly, lower for the 50% than for the 100% contrast (60.5 Hz vs 63.1 Hz; $F_{(1, 13)} = 10.2$, $p = 0.007$; Fig 6B).

**Rank-order consistency of gamma parameters.** The correlations between GR frequencies measured in different contrast conditions were generally very high for GR frequency (all R's > 0.89; all p's < 0.0001), with the noticeable exception of the static stimulus (S3 Table). For power, all the between-contrast correlations were modestly or highly reliable (Fig 7A).

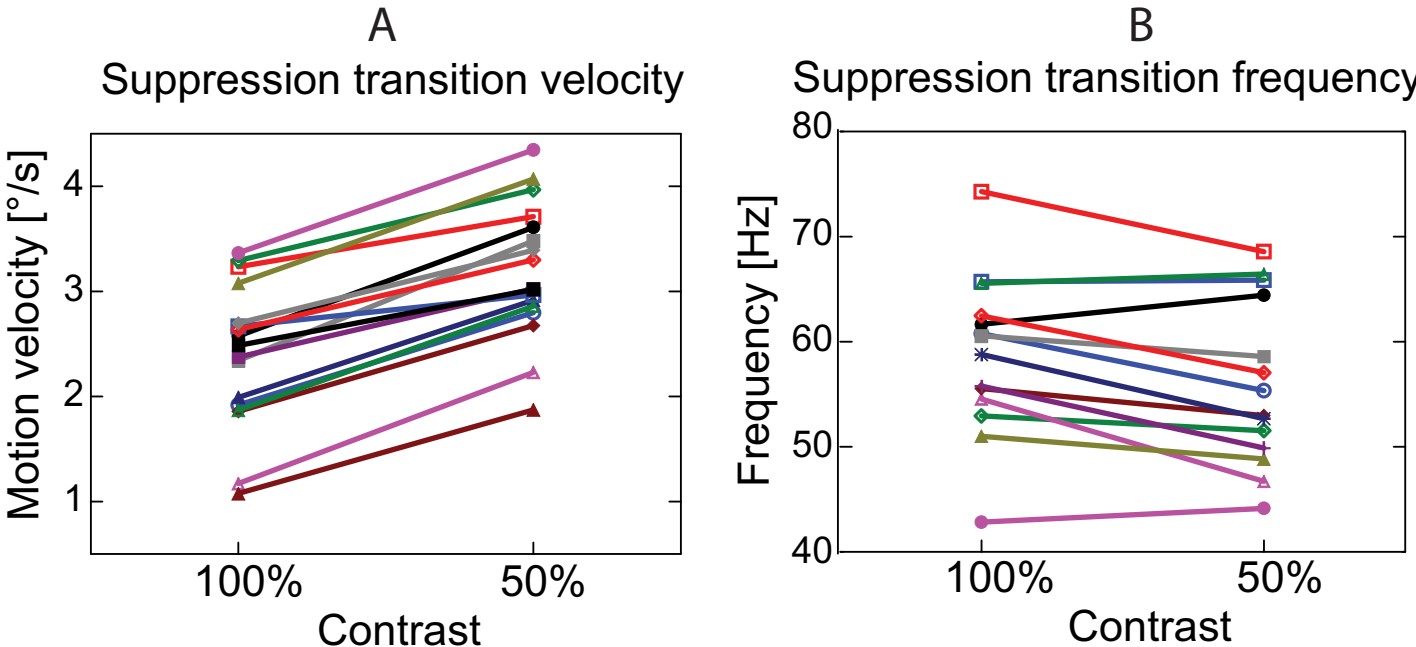

**Fig 6. Effect of contrast on the gamma suppression transition point: individual variability.** A. The centre of gravity of visual motion velocity that approximates the GR suppression transition point ('suppression transition velocity'—STVel). The STVel increased with decreasing contrast in all subjects (p<1e-6). B. The centre of gravity that approximates the peak frequency of the GR at the suppression transition point ('suppression transition frequency'—STFreq) in 14 subjects for whom frequency could be assessed in all experimental conditions (see Materials and Methods for details). The STFreq slightly, but significantly, decreased with decreasing the contrast (p = 0.007).

The correlations between velocity conditions were also generally high for frequency (S3 Table), but varied in case of power. The power of the GR elicited by static stimuli did not predict the power of the GR elicited by gratings moving with 'medium' or 'fast' velocities (Fig 7B). The results were very similar for the two contrasts (see S1 Fig for the both contrasts).

The correlation between STVel values measured at the 50% and 100% contrast was high (Fig 7C, STVel: $R_{(17)}$ = 0.91, p<1e-6), suggesting that this measure reliably characterises a subject's rank position in the group, irrespective of contrast changes.

To summarize, rank-order consistency for GR frequency was generally high across all conditions except for stationary stimuli. For GR power, a rank-order consistency between stationary and medium-to-fast moving gratings was lacking.

### Alpha-beta frequency range

**Localization and reliability.** Fig 8A shows group probability maps for source localization of the alpha-beta response to the moving visual gratings.

Clusters of significant (p<0.05) alpha-beta power suppression were found in nearly all subjects and conditions. For the 100% contrast stimuli, the clusters were absent in two subjects: in one subject in all velocity conditions and in yet another subject in the static condition. The suppression measured at the selection of the 'maximally suppressed' voxels was significant in all motion velocity conditions at 100% contrast. For the 50% contrast stimuli, a significant cluster for alpha-beta suppression was absent in only one subject and only in response to the static stimulus. In yet another subject, the alpha-beta suppression measured at the voxels' selection was not significant (p>0.0001) for the 1.2˚/s condition. The grand average spectra in the alpha-beta range for the maximally suppressed voxels are shown in Fig 8B.

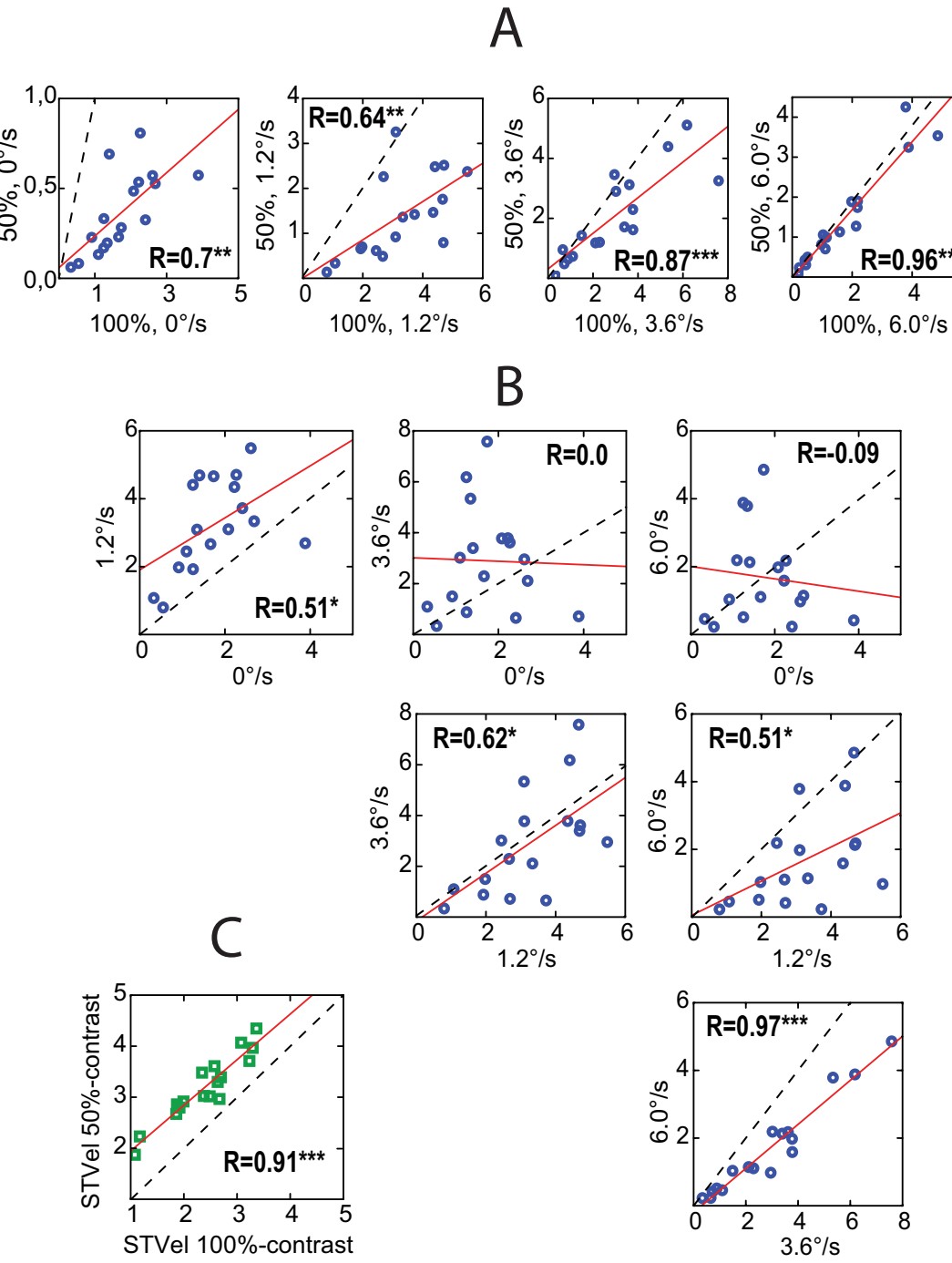

**Fig 7. Rank-order consistency of the GR power across conditions.** A. Correlations between GR power measured at 50% and 100% contrasts. B. Correlations between GR power values measured in different velocity conditions at 100% contrast (see S1 Fig in Supporting Information for a similar figure for the 50% contrast stimuli). Blue dots in A and B denote individual GR power values. C. Correlations between STVel values at the two contrasts. Green squares denote individual STVel values. The linear regression is shown in red. Dashed lines in all plots correspond to the axis of symmetry.

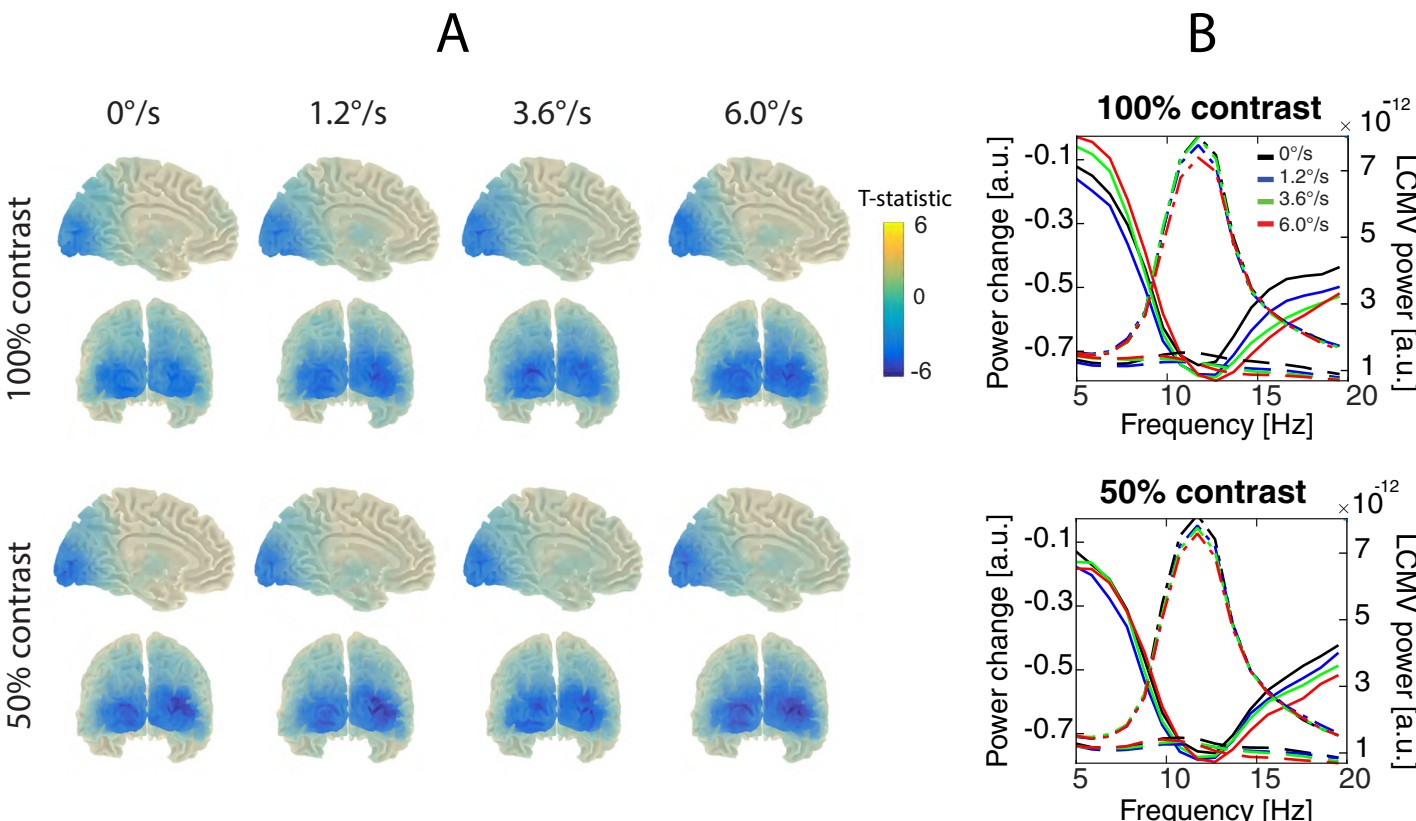

**Fig 8. Grand average power changes in the alpha-beta range.** A. Statistical maps of the stimulation-related changes in the two luminance contrasts and four motion velocity conditions. Blue color corresponds to suppression of alpha-beta power. B. Spectral changes at a selection of 'maximally suppressed' voxels. Dashed lines show absolute power spectra (right y-axis, arbitrary units) corresponding to the periods of visual stimulation. The dot -dashed lines of the same colors show the power spectra for the respective pre-stimulus intervals (also right y-axis). Solid lines show respective stimulation-related changes in power ([post-pre]/pre ratio, left y-axis, arbitrary units). Neither contrast nor velocity had significant effects on the magnitude of alpha-beta suppression.

ANOVA with factors Contrast and Velocity revealed neither significant main effects of Contrast and Velocity nor Contrast*Velocity interaction for the 'x', 'y', or 'z' coordinates of the 'maximally suppressed' voxel (all p's>0.15).

We then used ANOVA with factors Band (alpha-beta vs gamma), Contrast, Velocity to compare the positions of the voxels with a maximal induced power changes in gamma and alpha-beta frequency ranges. The effect of Band was highly significant for the absolute value of the 'x' coordinate ($F_{(1,16)} = 81.9$, p<1e-6). The maximally suppressed alpha-beta voxel was positioned substantially more laterally (on average 25 mm from the midline) than that of the maximally induced gamma voxel (on average 8 mm from the midline). As can be seen from S2A and S2B Table, the maximally suppressed alpha-beta voxel was more frequently located in the lateral surface of the occipital lobe than in the calcarine sulcus or cuneus region. No Band related differences in the y or z coordinates were found.

**Effects of contrast and velocity.** To check if the magnitude or peak frequency of the alpha-beta suppression response depends on the properties of the stimulation, we performed ANOVA with factors Contrast and Velocity. The peak frequency of the alpha-beta suppression sightly, but significantly, increased with increasing motion velocity (Velocity: $F_{(3,36)} = 7.1$, $\varepsilon = 0.57$, p<0.01; 11.8 Hz for 0°/s, 11.9 Hz for 1.2°/s, 12.0Hz for 3.6°/s, 12.2 Hz for 6.0°/s). There was no significant effects of Contrast or Velocity or their interaction effect for the alpha-beta suppression magnitude (all p's>0.5).

## Discussion

We tested for the combined effect of contrast and velocity of drifting visual gratings on the power of induced MEG gamma oscillations. For both full-contrast and low-contrast conditions, the GR power initially increased with increasing motion velocity and then decreased at yet higher velocities, which is consistent with a previous report [18]. Lowering contrast led to a highly reliable shift of the gamma suppression transition point to higher velocities. This finding points to an additive effect of contrast and velocity and suggests that intensive excitatory input plays a crucial role in GR suppression. The effects of contrast and velocity were furthermore specific for the GR power and were absent in the alpha-beta frequency range.

### The role of excitatory drive in induced gamma response

In line with the previous studies on humans and non-human primates [12, 14–16, 59], we found that the peak frequency of the visual GR increased with increasing luminance contrast (Fig 5A). Also in accordance with a number of previous reports [16, 18, 60–62], the GR peak frequency increased with increasing velocity of visual motion (Fig 5A). The increase in the GR peak frequency at high contrasts and fast velocities of visual motion is likely to be a consequence of a growth of tonic excitation of $I$-neurons that control gamma oscillation frequency [4].

The low-contrast stimuli, as compared with the high-contrast ones, induced GRs of lower power, which was especially evident at slower velocities (Fig 5B). The weaker GR power at lower contrast could be explained by a reduced number of neurons involved in gamma generation [17] and/or poor neural synchronization across distributed cortical populations [63]. The stronger extrastriate feedback signal to V1 produced by large high-contrast gratings [64] has been suggested to facilitate visual gamma synchrony at the macroscopic level [63]. Therefore, the low power of GR at low contrast could be partially explained by lower top-down modulatory input to V1.

The most important finding of our study is the effect of luminance contrast on the bell-shaped velocity-related changes in GR. Although at both low and high contrasts, the initially facilitative effect of increasing velocity was substituted by a suppressive one, this shift occurred at a higher velocity for the low contrast condition (Figs 3 and 6A). This contrast-dependency of the STVel is a robust phenomenon that we observed in all subjects that we tested (Fig 6A).

In general, the results support the 'excitatory drive' model, which predicts that GR suppression should occur at a certain high level of excitatory drive, which is achieved at a relatively higher velocity when contrast is reduced. Concurrently, our results do not support the 'velocity tuning' hypothesis. Animal studies have shown that at lower contrasts, the majority of responsive neurons in V1 [32, 47, 48], LGN [47], and even the retina [65, 66] are tuned to *lower* temporal frequencies. Therefore, if the maximal GR would correspond to an 'optimal' velocity, the GR suppression transition point would have shifted to a lower visual motion velocity when contrast is reduced.

**State and trait dependency of visual gamma parameters.** Although being mostly genetically determined [9] and highly reproducible across time [8, 62], visual gamma oscillations are strongly dependent on the properties of the visual stimulation. This raises the question of whether gamma measured in response to different visual stimuli—e.g. those having different contrasts or drifting with different rates—reflect the same or different neurophysiological traits.

Several studies that sought to find a link between visual gamma oscillations and processes of neural $E$ and $I$ in the human brain [67–70] analyzed GRs in a single experimental condition. Such an approach assumes that the features of the GR do not substantially vary with stimulus properties in terms of the rank-order consistency of the individual values. Recently, van Pelt

et al [16] reported high between-condition correlations for GR peak frequency and power measured at two contrasts (50% and 100%) and at three velocities (0, 0.33˚/s, 0.66˚/s) that support this view. However, the relatively slow velocities chosen by van Pelt et al are all likely to correspond to the ascending branch of the bell–shaped curve that characterizes the velocity-dependency of GR power. Using a broader range of velocities, we report a lack of correlations between the amplitudes of the GR caused by static and rapidly moving (3.2˚/s, 6.0˚/s) stimuli (Fig 7B). This suggests that the strength of the GR at the ascending and descending branches of the bell-curve reflects different neural traits and is mediated by distinct neural processes that remain largely unknown.

In the context of continuing the search for the functional relevance of visual gamma oscillations, it is important that there was a remarkably strong and highly reliable correlation between individuals' GR suppression transition velocities at the high and low contrasts (Spearman $R_{(17)}$ = 0.91). This points to a high within-subject reliability of individual STVel values across a range of experimental conditions. As we discuss in the following section, this relational gamma-based measure may reflect the regulation of the *E-I* balance in the visual cortex.

## GR suppression transition point and E-I balance in the visual cortex

Active states of the brain associated with processing of sensory information are usually characterized by a relative predominance of high-frequency oscillations and relative suppression of low-frequency oscillations [49]. In light of this general phenomenon, the suppression of the GR at high levels of input drive (high contrast and fast velocity) can appear counterintuitive. However, some modeling studies do predict that gamma synchrony should cease at a sufficiently high level of excitatory drive where *I*-neurons are excessively excited [43, 45]. The resulting *asynchronous* activity of the highly excited *I* neurons is particularly effective in down-regulating activity in the excitatory principle cells [43], and therefore play an important role in homeostatic regulation of the neural *E-I* balance. Following this line of reasoning, the transition to suppression of the gamma response and/or shallower slope of gamma suppression at a relatively higher level of excitatory drive could reflect less effective inhibition. In indirect support of this assumption, we have recently described a lack of velocity-related gamma suppression in a subject with epilepsy and occipital spikes [18]. A link between reduced velocity-related GR suppression and sensory hypersensitivity [71] further supports this conjecture.

The shape of the GR modulation curve and, in particular, the 'suppression transition velocity' parameter evaluated in the present study, may provide important information about regulation of the *E-I* balance in the visual cortex. Future studies in clinical populations characterized by elevated cortical excitability would help to assess the potential value of GR suppression as a biomarker for impaired regulation of the *E-I* balance. In particular, visual gamma oscillations are becoming a popular subject of research as a potential biomarker of an altered *E-I* balance in neuropsychiatric disorders, such as schizophrenia and autism spectrum disorders [7, 72–75]. In a majority of these clinically oriented studies, the parameters of gamma oscillations (power, frequency) were investigated in a single experimental condition. Our present and previous [18, 71, 76] results suggest that *changes* of gamma parameters caused by changes in sensory input intensity may be particularly informative for detecting *E-I* balance abnormalities in neuro-psychiatric disorders.

## Magnitude of the alpha-beta suppression is not modulated by excitatory drive

The contrast/velocity-related changes in alpha-beta power were clearly different from the effects in the gamma range. *First*, whereas the maximum of the GR was localized near the

calcarine sulcus, the alpha-beta suppression occurred in more lateral areas of the occipital lobes (Fig 2 vs. Fig 8; see also S1 and S2 Tables for localization of the maximal effect voxels). This finding agrees with other studies that found more lateral changes in the alpha-beta than in the gamma frequency range [8, 77, 78]. *Second*, neither the contrast nor the velocity of the gratings affected power in the alpha-beta frequency range. A lack of such effects conflicts with the idea that variation in alpha power can be used as an index of the excitatory state of the visual cortex. On the other hand, our findings do not contradict the numerous results associating alpha-band activity with attention-related processes [79–82].

### Limitations of the study and directions for future research

Limitations in this study provide motivation for future studies. *Firstly*, although our findings favor 'the excitatory drive' explanation of the bell-shaped dependency of gamma oscillations power on visual motion velocity, further studies are needed to prove this hypothesis. In particular, the role of the *E-I* balance in regulating MEG GSS have to be verified using other methods, e.g. via manipulating cortical excitability with transcranial direct current stimulation (tDCS), transcranial magnetic stimulation (TMS), or pharmacological drugs while measuring GSS before and after intervention. Invasive experiments in animals would also provide a robust validity check of this hypothesis. Computer modeling could further contribute to understanding neural mechanisms underlying changes in gamma parameters observed in the present study. *Secondly*, although the effect of contrast on the 'suppression transition point' was very reliable (17 of 17 subjects), replication of such findings in another sample would be desirable.

### Conclusions

Our findings provide strong support for the hypothesis linking velocity- and contrast-related attenuation of gamma response to the strength of excitatory drive. At the same time, they indicate that 'velocity tuning' of V1 neurons does not play a primary role in regulating the magnitude of the gamma response.

We anticipate that suppression transition velocity (or related measures) may appear useful to reveal E-I dysfunctions in brain disorders. Apart from having theoretical relevance, this index evades several of the limitations of peak gamma amplitude and frequency parameters. Firstly, it is inherently relational, and is therefore less sensitive to inter-individual differences in cortical anatomy and SNR. Secondly, it circumvents the ambiguity related to single-condition assessments of gamma amplitudes, which do not always correlate with each other (e.g., when measured with static vs. 'fast velocity' conditions). Thirdly, its individual values have remarkable rank-order consistency when measured at different contrasts, and therefore should have high test–retest stability. Altogether, these considerations suggest that the gamma suppression transition velocity has a translational potential as an index of the E-I balance for informing clinical practice and trials.

### Supporting information

**S1 Fig. Supplementary Fig 1.** Correlations between GR power values measured at different velocities in 100% and 50% contrast conditions.
(PDF)

**S2 Fig. Supplementary Fig 2.** Distributions of individual peak frequencies of gamma responses induced by 50% and 100% contrast gratings moving at different velocities.
(PDF)

**S1 File. Supplementary Dataset 1.** Gamma parameters in the source space.
(XLS)

**S2 File. Supplementary Dataset 2.** Alpha-beta parameters in the source space.
(XLS)

**S1 Table. Supplementary Table 1.** Position of the voxel with maximal gamma response in the 'static', 'slow', 'medium', and 'fast' velocity conditions in each of the 17 participants.
(PDF)

**S2 Table. Supplementary Table 2.** Position of the voxel with maximal alpha-beta suppression in the 'static', 'slow', 'medium' and 'fast' velocity conditions in each of the 17 participants.
(PDF)

**S3 Table. Supplementary Table 3.** Partial correlations between the peak frequencies of GRs elicited by visual gratings drifting with different velocities in the two contrast conditions.
(PDF)

## Acknowledgments

We thank all volunteers who participated in our experiment.

## Author Contributions

**Conceptualization:** Elena V. Orekhova, Tatiana A. Stroganova.

**Data curation:** Andrey O. Prokofyev, Anastasia Yu. Nikolaeva.

**Formal analysis:** Elena V. Orekhova, Andrey O. Prokofyev.

**Funding acquisition:** Justin F. Schneiderman, Tatiana A. Stroganova.

**Methodology:** Elena V. Orekhova.

**Project administration:** Tatiana A. Stroganova.

**Software:** Elena V. Orekhova.

**Supervision:** Tatiana A. Stroganova.

**Visualization:** Elena V. Orekhova.

**Writing – original draft:** Elena V. Orekhova.

**Writing – review & editing:** Elena V. Orekhova, Andrey O. Prokofyev, Anastasia Yu. Nikolaeva, Justin F. Schneiderman, Tatiana A. Stroganova.

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
