## [Decision Letter · Decision Letter 0]

9 Dec 2019

PONE-D-19-27722

Additive effect of contrast and velocity suggests the role of strong excitatory drive in suppression of visual gamma response

PLOS ONE

Dear Dr Orekhova,

Thank you for submitting your manuscript to PLOS ONE. After careful consideration, we feel that it has merit but does not fully meet PLOS ONE’s publication criteria as it currently stands. Therefore, we invite you to submit a revised version of the manuscript that addresses the points raised during the review process.

The manuscript has been evaluated by three reviewers, and their comments are available below.

The reviewers are positive about this work but have raised a number of concerns that need attention. They request additional contextualization and have raised concerns regarding the overall presentation and the specific hypotheses, and request clarification regarding the statistical analyses. 

Could you please revise the manuscript to carefully address the concerns raised?

We would appreciate receiving your revised manuscript by Jan 20 2020 11:59PM. To enhance the reproducibility of your results, we recommend that if applicable you deposit your laboratory protocols in protocols.io, where a protocol can be assigned its own identifier (DOI) such that it can be cited independently in the future. For instructions see: http://journals.plos.org/plosone/s/submission-guidelines#loc-laboratory-protocols

 A rebuttal letter that responds to each point raised by the academic editor and reviewer(s). This letter should be uploaded as separate file and labeled 'Response to Reviewers'. A marked-up copy of your manuscript that highlights changes made to the original version. This file should be uploaded as separate file and labeled 'Revised Manuscript with Track Changes'. An unmarked version of your revised paper without tracked changes. This file should be uploaded as separate file and labeled 'Manuscript'.

We look forward to receiving your revised manuscript.

Kind regards,

Vanessa Carels

Staff Editor

PLOS ONE

Journal Requirements:

1.

2. 

We note that you have indicated that data from this study are available upon request. PLOS only allows data to be available upon request if there are legal or ethical restrictions on sharing data publicly. For more information on unacceptable data access restrictions, please see http://journals.plos.org/plosone/s/data-availability#loc-unacceptable-data-access-restrictions.

534 This work was supported by the Moscow State University of Psychology and Education; the

535 Charity Foundation "Way Out"; Swedish Childhood Cancer (# MT2014-0007); Knut and Alice

536 Wallenberg Foundation (#2014.0102) and Swedish Research Council (# 2017-0068).

Please remove any funding-related text from the manuscript and let us know how you would like to update your Funding Statement. Currently, your Funding Statement reads as follows:  "The funders had no role in study design, data collection and analysis, decision to publish, or preparation of the manuscript."

Reviewers' comments:

Reviewer's Responses to Questions

**Comments to the Author**

1. Is the manuscript technically sound, and do the data support the conclusions?

Reviewer #1: Yes

Reviewer #2: Yes

Reviewer #3: Yes

2. Has the statistical analysis been performed appropriately and rigorously? 

Reviewer #1: Yes

Reviewer #2: Yes

Reviewer #3: I Don't Know

3. Have the authors made all data underlying the findings in their manuscript fully available?

Reviewer #1: Yes

Reviewer #2: No

Reviewer #3: Yes

4. Is the manuscript presented in an intelligible fashion and written in standard English?

Reviewer #1: Yes

Reviewer #2: Yes

Reviewer #3: Yes

5. Review Comments to the Author

Reviewer #1: The paper is excellent and clearly written and presented throughout. Gamma oscillations are increasingly found to differ in clinical groups and a clear study on the effects of stimulus parameters on gamma is welcome.

Reviewer #2: The article describes the results of an experiment with humans in which the authors present dynamics/static visual gradings with two levels of contrast to subjects and collect the local cortical oscillations through the MEG technique. They observed how changing the velocity/contrast of the visual gradings affects the power and frequency of evoked gamma-frequency responses (GR) in the visual cortex. They look for a bell-shaped response of GR power with increasing grading velocity, which is presumably caused by input induced changes in the excitatory-inhibitory balance of the local network. The core outcome of the work is to show that changing the contrast of the visual gradings modifies the peak frequency of the bell shape response (which they named GR suppression transition point). To explain the results, they list two hypotheses. In one, the "velocity tuning," the bell curve would be skewed to lower frequencies or remain unaltered with added contrast. The alternative, the "excitatory drive," predicts that the bell curve would skew to higher frequencies as contrast increases. The article claims to have evidence for the last.

The results and experimental methods are sound and interesting. However, the article requires significant changes to increase readability, and some clarification regards some aspects of the analysis before publication. I would thus recommend "major revisions," as I feel that the text needs considerable work, but I would not complain to include it as "minor" as the experimental part seems to be almost ready.

The main weakness of the text is the "two hypotheses" presented to justify the study. As a computational modeler, I am very kin to articles with this format: hypothesis(es) / test/ discussion. However, I am not convinced about the power of these two hypotheses presented to explain the observed results in this particular study. My impression is that the two hypotheses are not justified. By experience, I know that E-I interactions frequently produce unpredictable outcomes. Moreover, it is oversimplistic to ensure that only one of the two mechanisms is at work. Thus, in the article, the reader faces two strong predictions that seem to be supported only by intuition. The keep the article in such a format, the authors should either describe in detail the mechanisms by which the "velocity tuning" or "excitatory drive" produces the outcomes they predict (with new text and diagrams) or simulate the outcome. Alternatively, the authors should tone down the article into a more explorative piece and only mention the hypotheses in the discussion. In such a case, the authors should also modify the title, abstract, and results, as in many cases, the hypotheses are used to justify the actions. For example, line 313, "as predicted by the excitatory drive model."

In regards to the analysis, the projector has a refresh rate of 60 Hz, which sits the induced stimuli well within the gamma frequency range. The authors need to discuss the possible implications for the results. Is this accounted for in the analysis?

Some small comments:

In the Abstract, the authors talk about GR enhancement, attenuation, and strength. It should be clear that this is about GR power.

Line 34: the sentence can be more formal.

Authors briefly discuss the effect of E-I in gamma frequency oscillations citing mostly experimental papers, and latter discuss some model in very abstract form. There are many essential papers about models of gamma frequency oscillations considering feedback and feedforward inhibition and E-I balance that should also be mentioned.

Line 372: provability?

Figure 5: can indicate the significance of the differences?

Reviewer #3: Thank you for asking me to review this manuscript, I recommend that it is accepted with minor revisions.

The authors investigated the effect of velocity and contrast of annular gratings in inducing a visual gamma response. Specifically they sought to explain why velocity, up to a certain point, increases the power of gamma response and beyond that point decreases it. This is an important question as features of gamma oscillations are increasingly being proposed as robust biomarkers in neuropsychiatric disorders and therefore are of interest to neuroscientists as well as clinical researchers in the fields of psychiatry and neurology.

The authors proposed two plausible yet incompatible hypotheses: (1) the velocity tuning model and (2) the excitatory drive model. Both these models could be theoretically correct, the authors therefore tested them experimentally in an elegant study. They concluded that the excitatory drive model was supported and that this finding was specific the gamma range of frequencies.

Although I am not an expert in MEG methodology, the description of methods appears detailed and robust. Likewise, their statistical analysis seems appropriate but this is not my field of expertise. Their conclusions are justified by the results.

I would like to raise a some minor concerns:

1. The study protocol and hypotheses do not appear to have been pre-registered. Had this been done, we could have more confidence in the study's findings. Without pre-registration, the study should be considered exploratory and needs replication in a separate sample.

2. I am surprised the authors did not mention the translatable potential of gamma oscillations in psychotic disorders such as Schizophrenia. Gamma-based biomarkers have enormous potential in the prediction and risk stratification of psychosis and as a biomarker for novel treatments.

3. The authors do not seem to have mentioned any limitations of their study. For example, I expected them to mention this is a relatively small sample and requires replication.

4. The authors do not provide detailed description of how their sample was recruited or their characteristics.

5. Line 81 - the reference is not cited numerically.

6. The Results section seems longwinded, containing passages that would be more appropriately included in Methods or Discussion. For example, Lines 245-246, Methods; Lines 308-313 Discussion; Lines 315-351 mostly Methods with some Results.

7. There are some typos of symbols in the following lines: 278, 288, 404.

8. The conclusions are long-winded. Lines 514-520 would be more appropriately included in the discussion section.

6. PLOS authors have the option to publish the peer review history of their article (what does this mean?). If published, this will include your full peer review and any attached files.

Reviewer #1: Yes: Dr Myles Jones

Reviewer #2: No

Reviewer #3: Yes: Dr Thomas Reilly

---

## [Author Response · Author response to Decision Letter 0]

22 Jan 2020

Reviewers' comments:

Reviewer #1: ‘The paper is excellent and clearly written and presented throughout. Gamma oscillations are increasingly found to differ in clinical groups and a clear study on the effects of stimulus parameters on gamma is welcome.’

We are grateful to the reviewer for a positive assessment of our work.

Reviewer #2: ‘The article describes the results of an experiment with humans in which the authors present dynamics/static visual gradings with two levels of contrast to subjects and collect the local cortical oscillations through the MEG technique. They observed how changing the velocity/contrast of the visual gradings affects the power and frequency of evoked gamma-frequency responses (GR) in the visual cortex. They look for a bell-shaped response of GR power with increasing grading velocity, which is presumably caused by input induced changes in the excitatory-inhibitory balance of the local network. The core outcome of the work is to show that changing the contrast of the visual gradings modifies the peak frequency of the bell shape response (which they named GR suppression transition point). To explain the results, they list two hypotheses. In one, the "velocity tuning," the bell curve would be skewed to lower frequencies or remain unaltered with added contrast. The alternative, the "excitatory drive," predicts that the bell curve would skew to higher frequencies as contrast increases. The article claims to have evidence for the last

The results and experimental methods are sound and interesting. However, the article requires significant changes to increase readability, and some clarification regards some aspects of the analysis before publication. I would thus recommend "major revisions," as I feel that the text needs considerable work, but I would not complain to include it as "minor" as the experimental part seems to be almost ready.

The main weakness of the text is the "two hypotheses" presented to justify the study. As a computational modeler, I am very kin to articles with this format: hypothesis(es) / test/ discussion. However, I am not convinced about the power of these two hypotheses presented to explain the observed results in this particular study. My impression is that the two hypotheses are not justified. By experience, I know that E-I interactions frequently produce unpredictable outcomes. Moreover, it is oversimplistic to ensure that only one of the two mechanisms is at work. Thus, in the article, the reader faces two strong predictions that seem to be supported only by intuition. The keep the article in such a format, the authors should either describe in detail the mechanisms by which the "velocity tuning" or "excitatory drive" produces the outcomes they predict (with new text and diagrams) or simulate the outcome.’

We thank the Reviewer for these comments. In the Introduction we now provide more detailed justification of these two hypotheses as well as the alternative predictions they generate. Each of the two hypotheses lead to strong physiologically-based predictions regarding changes in the “optimal velocity” produced by a decrease in contrast. However, it is worth emphasizing that the detailed neural mechanisms of the predicted outcomes remain largely unknown. 

The ‘tuning’ hypothesis can be characterized as an obvious one for neurophysiologists as it is used to describe a variety of other phenomena. Indeed, tuning curves provide the first-order description of virtually every sensory system, from orientation columns in the vertebrate visual cortex, to place cells in the hippocampus and wind-detecting neurons in the cricket cercal system (Abbott and Dayan, 2001). Due to its ubiquitous application and straightforward formulation, the “tuning” has been previously used to explain the bell-shaped dependencies on temporal frequencies in EEG (Fawcett et al., 2004) and fMRI (Singh et al., 2000). Other alternatives have not been even considered by the authors cited above. Meanwhile, there is a way to test the “velocity tuning” hypothesis experimentally. In terms of firing rate, the majority of neurons in V1 are ‘tuned’ to specific temporal frequency/velocity of visual stimulation. If number of the ‘tuned’ neurons is the main factor that affects velocity-related changes in the GR power, the ‘optimal’ stimuli that activate more neurons should cause strongest gamma activity. From this point of view, changing gratings’ velocity should similarly affect gamma oscillations and multi-unit spiking activity (MUA). Fortunately, the effect of contrast on “velocity tuning” has been described for MUA in the visual cortex of many mammals including non-human primates. In all these species a decrease in the stimulus contrast shifted the tuning of the motion-sensitive V1 neurons to lower temporal (Alitto and Usrey, 2004; Livingstone and Conway, 2007; Priebe et al., 2006). Thus, the “velocity tuning” hypothesis predicts that the same effect of contrast will be observed for gamma tuning curve in human V1. When we tested this prediction, the result was the opposite of the predicted. This discrepancy between MUA and gamma response is not an isolated finding; it has been previously described in V1 of non-human primates (Gieselmann and Thiele, 2008; Salelkar et al., 2018). 

The other important factor that affects GR power is synchronization between E and I neurons (e.g. (Buzsaki and Wang, 2012)). In turn, neural synchronization in the gamma range is highly sensitive to E-I balance in the whole network. Our alternative “E-I balance” explanation for velocity-dependency of the GR power is grounded in:

i) the modelling results that predicted that increasing input strength, if accompanied by strong decrease in the E/I ratio, should lead to the bell-shaped changes in gamma synchronization, i.e. in the GR power (e.g. in Cannon, McCarthy, Lee, Lee, Borgers, Whittington and Kopell, European Journal of Neuroscience, Vol. 39, pp. 705–719, 2014); 

ii) the recent experimental findings that did show unbalanced changes in E and I synaptic conductance in V1 produced by increasing intensity of visual stimulation (Adesnik, 2017); 

iii) the experimental ‘in vitro’ studies that demonstrated the predicted negative dependency of gamma oscillations power on the disproportional rise in inhibition strength (Mann and Mody, 2010; Towers et al., 2004). 

If the maximal GR corresponds to an optimal E/I ratio, it is likely that at lower contrast this ‘optimal’ ratio will be achieved at higher velocity. This is exactly what we found.

We fully agree with the reviewer that using MEG we cannot uncover mechanisms of nonlinear changes in gamma power with increasing stimulation intensity. We now clearly state this as a limitation of the study in the Discussion. Nevertheless, we believe that our physiologically-based explanation is plausible and can be tested in future animal and computer modelling studies.

‘Alternatively, the authors should tone down the article into a more explorative piece and only mention the hypotheses in the discussion. In such a case, the authors should also modify the title, abstract, and results, as in many cases, the hypotheses are used to justify the actions. ‘

We hope that the information added to the Introduction now better justifies our study. 

‘For example, line 313, "as predicted by the excitatory drive model."’

As suggested by Reviever#3 we moved a part of the text in the Discussion section.

‘In regards to the analysis, the projector has a refresh rate of 60 Hz, which sits the induced stimuli well within the gamma frequency range. The authors need to discuss the possible implications for the results. Is this accounted for in the analysis?’

We thank the Reviewer for this question. Yes, we are well aware about possible contamination from the 60 Hz projector refresh rate (photic driving). The ‘suspicious’ 60 Hz spectral peaks were sometimes present in our previous study in children (Orekhova et al., 2018b). Subtraction of the evoked response usually reduced or eliminated these peaks. In our adult sample such photic-driving peaks were not visually detectable. Nevertheless, to prevent/reduce possible 60 Hz contamination we subtracted the averaged responses from the raw data, as it is mentioned in the Methods section. Now we explain more explicitly why we did this:

line 240: ‘In order to decrease the contribution of phase-locked activity related to appearance of the stimulus on the screen and 60 Hz refresh rate of the projector, we subtracted the average evoked response from each data epoch. ’ 

We also plotted histograms of peak frequency distribution (S2 Fig in Supporting Materials). As it can be seen from these histograms, the peak frequencies do not seem to cluster at 60 Hz. 

‘Some small comments:

In the Abstract, the authors talk about GR enhancement, attenuation, and strength. It should be clear that this is about GR power.’

We corrected the abstract.

‘Line 34: the sentence can be more formal.’

We corrected this sentence.

Previously: 

Second, the GR attenuation at high velocities of visual motion can be caused by ‘too strong’ excitatory drive that disrupts gamma synchrony via stronger growth of inhibition than excitation. 

Now:

Second, the GR power attenuation at high velocities of visual motion can be caused by changes in excitation/inhibition balance with increasing excitatory drive.

‘Authors briefly discuss the effect of E-I in gamma frequency oscillations citing mostly experimental papers, and latter discuss some model in very abstract form. There are many essential papers about models of gamma frequency oscillations considering feedback and feedforward inhibition and E-I balance that should also be mentioned.’

Unfortunately, depending on neural architecture and parameters used, different models produce different predictions regarding input-related changes in gamma parameters that are not always compatible with the experimental results. As an example, model used in the study of Jia et al (Jia et al., 2013)failed to describe clearly nonlinear contrast-dependent changes in GR power observed in their data (Fig. 2c). Since our study is experimental, we cited the model/hypothesis that, as we think, is most compatible with our findings. We do not feel that discussing other less relevant models will help understanding of our experimental results.

‘Line 372: provability?’

Thank you, this has been changed to ‘probability’.

‘Figure 5: can indicate the significance of the differences?’

We now added the p-values to the figure.

REFERENCES

Abbott, L.F., Dayan, P., 2001. Theoretical Neuroscience. Computational and Mathematical Modeling of Neural Systems MIT Press, Cambridge (Massachusetts).

Adesnik, H., 2017. Synaptic Mechanisms of Feature Coding in the Visual Cortex of Awake Mice. Neuron 95, 1147–1159.

Alitto, H.J., Usrey, W.M., 2004. Influence of contrast on orientation and temporal frequency tuning in ferret primary visual cortex. J Neurophysiol 91, 2797-2808.

Brealy, J.A., Shaw, A., Richardson, H., Singh, K.D., Muthukumaraswamy, S.D., Keedwell, P.A., 2015. Increased visual gamma power in schizoaffective bipolar disorder. Psychological Medicine 45, 783-794.

Buzsaki, G., Wang, X.J., 2012. Mechanisms of Gamma Oscillations. Annual Review of Neuroscience, Vol 35 35, 203-225.

Dickinson, A., Bruyns-Haylett, M., Smith, R., Jones, M., Milne, E., 2016. Superior Orientation Discrimination and Increased Peak Gamma Frequency in Autism Spectrum Conditions. Journal of Abnormal Psychology 125, 412-422.

Fawcett, I.P., Barnes, G.R., Hillebrand, A., Singh, K.D., 2004. The temporal frequency tuning of human visual cortex investigated using synthetic aperture magnetometry. Neuroimage 21, 1542-1553.

Gieselmann, M.A., Thiele, A., 2008. Comparison of spatial integration and surround suppression characteristics in spiking activity and the local field potential in macaque V1. Eur J Neurosci 28, 447-459.

Grent-'t-Jong, T., Rivolta, D., Sauer, A., Grube, M., Singer, W., Wibral, M., Uhlhaas, P.J., 2016. MEG-measured visually induced gamma-band oscillations in chronic schizophrenia: Evidence for impaired generation of rhythmic activity in ventral stream regions. Schizophrenia Research 176, 177-185.

Jia, X.X., Xing, D.J., Kohn, A., 2013. No Consistent Relationship between Gamma Power and Peak Frequency in Macaque Primary Visual Cortex. J Neurosci 33, 17-U421.

Levin, A.R., Nelson, C.A., 2015. Inhibition-Based Biomarkers for Autism Spectrum Disorder. Neurotherapeutics 12, 546-552.

Livingstone, M.S., Conway, B.R., 2007. Contrast affects speed tuning, space-time slant, and receptive-field organization of simple cells in macaque V1. J Neurophysiol 97, 849-857.

Mann, E.O., Mody, I., 2010. Control of hippocampal gamma oscillation frequency by tonic inhibition and excitation of interneurons. Nature Neuroscience 13, 205-U290.

Orekhova, E.V., Rostovtseva, E.N., Manyukhina, V.O., Prokofiev, A.O., Obukhova, T.S., Nikolaeva, A.Y., Schneiderman, J.F., Stroganova, T.A., 2019. Spatial suppression in visual motion perception is driven by inhibition: evidence from MEG gamma oscillations. bioRxiv.

Orekhova, E.V., Stroganova, T.A., Schneiderman, J.F., Lundstrom, S., Riaz, B., Sarovic, D., Sysoeva, O.V., Brant, G., Gillberg, C., Hadjikhani, N., 2018a. Neural gain control measured through cortical gamma oscillations is associated with sensory sensitivity. Hum Brain Mapp.

Orekhova, E.V., Sysoeva, O.V., Schneiderman, J.F., Lundstrom, S., Galuta, I.A., Goiaeva, D.E., Prokofyev, A.O., Riaz, B., Keeler, C., Hadjikhani, N., Gillberg, C., Stroganova, T.A., 2018b. Input-dependent modulation of MEG gamma oscillations reflects gain control in the visual cortex. Sci Rep 8, 8451.

Priebe, N.J., Lisberger, S.G., Movshon, J.A., 2006. Tuning for spatiotemporal frequency and speed in directionally selective neurons of macaque striate cortex. J Neurosci 26, 2941-2950.

Salelkar, S., Somasekhar, G.M., Ray, S., 2018. Distinct frequency bands in the local field potential are differently tuned to stimulus drift rate. J Neurophysiol.

Shaw, A.D., Knight, L., Freeman, T.C.A., Williams, G.M., Moran, R.J., Friston, K.J., Walters, J.T.R., Singh, K.D., 2019. Oscillatory, Computational, and Behavioral Evidence for Impaired GABAergic Inhibition in Schizophrenia. Schizophr Bull.

Singh, K.D., Smith, A.T., Greenlee, M.W., 2000. Spatiotemporal frequency and direction sensitivities of human visual areas measured using fMRI. Neuroimage 12, 550-564.

Towers, S.K., Gloveli, T., Traub, R.D., Driver, J.E., Engel, D., Fradley, R., Rosahl, T.W., Maubach, K., Buhl, E.H., Whittington, M.A., 2004. alpha 5 subunit-containing GABA(A) receptors affect the dynamic range of mouse hippocampal kainate-induced gamma frequency oscillations in vitro. Journal of Physiology-London 559, 721-728.

Reviewer #3: ‘Thank you for asking me to review this manuscript, I recommend that it is accepted with minor revisions.

The authors investigated the effect of velocity and contrast of annular gratings in inducing a visual gamma response. Specifically they sought to explain why velocity, up to a certain point, increases the power of gamma response and beyond that point decreases it. This is an important question as features of gamma oscillations are increasingly being proposed as robust biomarkers in neuropsychiatric disorders and therefore are of interest to neuroscientists as well as clinical researchers in the fields of psychiatry and neurology.

The authors proposed two plausible yet incompatible hypotheses: (1) the velocity tuning model and (2) the excitatory drive model. Both these models could be theoretically correct, the authors therefore tested them experimentally in an elegant study. They concluded that the excitatory drive model was supported and that this finding was specific the gamma range of frequencies.

Although I am not an expert in MEG methodology, the description of methods appears detailed and robust. Likewise, their statistical analysis seems appropriate but this is not my field of expertise. Their conclusions are justified by the results.’

We thank the Reviewer for these positive comments on our work.

‘I would like to raise a some minor concerns:

1. The study protocol and hypotheses do not appear to have been pre-registered. Had this been done, we could have more confidence in the study's findings. Without pre-registration, the study should be considered exploratory and needs replication in a separate sample.’

The effect (shift in the ‘suppression transition point’ to higher velocity with lowering contrast) was observed in all our subjects. So, it seems to be highly reliable. However, we agree that replication of this study would be desirable, and add this note to the limitations of the study.

‘2. I am surprised the authors did not mention the translatable potential of gamma oscillations in psychotic disorders such as Schizophrenia. Gamma-based biomarkers have enormous potential in the prediction and risk stratification of psychosis and as a biomarker for novel treatments.’

Thank you for this comment. We now include this note in the discussion.

Line 515:

 ‘The shape of the GR modulation curve and, in particular, the ‘suppression transition velocity’ parameter evaluated in the present study, may provide important information about regulation of the E-I balance in the visual cortex. Future studies in clinical populations characterized by elevated cortical excitability would help to assess the potential value of GR suppression as a biomarker for impaired regulation of the E-I balance. In particular, visual gamma oscillations are becoming a popular subject of research as a potential biomarker of an altered E-I balance in neuropsychiatric disorders, such as schizophrenia and autism spectrum disorders (Brealy et al., 2015; Dickinson et al., 2016; Grent-'t-Jong et al., 2016; Levin and Nelson, 2015; Shaw et al., 2019). In a majority of these clinically oriented studies, the parameters of gamma oscillations (power, frequency) were investigated in a single experimental condition. Our present and previous (Orekhova et al., 2019; Orekhova et al., 2018a; Orekhova et al., 2018b) results suggest that changes of gamma parameters caused by changes in sensory input intensity may be particularly informative for detecting E-I balance abnormalities in neuro-psychiatric disorders.’

‘3. The authors do not seem to have mentioned any limitations of their study. For example, I expected them to mention this is a relatively small sample and requires replication.’

We now included ‘limitation section’ in the discussion.

‘4. The authors do not provide detailed description of how their sample was recruited or their characteristics.’

We now added more details about the sample:

‘Seventeen neurologically healthy subjects (age 18-39, mean=27.2, sd=6.1; 7 males) were recruited for the study by advertising among students and stuff of the Moscow State University of Psychology and Education (MSUPE). All participants had normal or corrected to normal vision. None of the participants reported presence of psychiatric problems.’

‘5. Line 81 - the reference is not cited numerically.’

We corrected the reference.

‘6. The Results section seems longwinded, containing passages that would be more appropriately included in Methods or Discussion. For example, Lines 245-246, Methods; Lines 308-313 Discussion; Lines 315-351 mostly Methods with some Results.’

These parts of the ‘Results’ were moved to the appropriate sections. 

‘7. There are some typos of symbols in the following lines: 278, 288, 404.’

Thank you for noticing this. We corrected these typos.

‘8. The conclusions are long-winded. Lines 514-520 would be more appropriately included in the discussion section.’

We shortened the Conclusions.

---

## [Editor Report · Decision Letter 1]

28 Jan 2020

Additive effect of contrast and velocity suggests the role of strong excitatory drive in suppression of visual gamma response

PONE-D-19-27722R1

Dear Dr. Orekhova,

We are pleased to inform you that your manuscript has been judged scientifically suitable for publication and will be formally accepted for publication once it complies with all outstanding technical requirements.

With kind regards,

César Rennó‐Costa

Guest Editor

PLOS ONE

Additional Editor Comments (optional):

As a disclaimer, I've been assigned as a Guest Editor to this submission during the review process. I participated as a reviewer for the initial evaluation of this manuscript (Reviewer #2).

Considering the two positive reviews to the first submission and the reasonable response to my comments and to the comments of Reviewer #3, I consider the article ready for publication. 

Reviewers' comments:

Reviewers #1 and #3 were not consulted during this round of review. 

I have, still, two minor comments that are OPTIONAL to the authors: 

Line (90): The call for <Salekhar et al> is three sentences away to the reference number (19). I suggest to advance the reference to the first sentence.

Line (99): I might be worth to mention that a mechanism for the change in E/I balance with increased E in favor of stronger I response, as reported in this paragraph, is feedforward inhibition (https://onlinelibrary.wiley.com/doi/abs/10.1002/hipo.23093). As a disclaimer, I'm co-author on this study. I did not mention it during my review process to avoid a possible conflict of interest before an editorial decision was made.

---

## [Editor Report · Acceptance letter]

5 Feb 2020

PONE-D-19-27722R1 

Additive effect of contrast and velocity suggests the role of strong excitatory drive in suppression of visual gamma response 

Dear Dr. Orekhova:

I am pleased to inform you that your manuscript has been deemed suitable for publication in PLOS ONE. Congratulations! Your manuscript is now with our production department. 

With kind regards,

on behalf of

Dr. César Rennó‐Costa 

Guest Editor

PLOS ONE